# Antimicrobial resistance in *Campylobacter jejuni* and *Campylobacter coli* isolated from small poultry flocks in Ontario, Canada: A two-year surveillance study

Csaba Varga[1]*, Michele T. Guerin[2], Marina L. Brash[3], Durda Slavic[3], Patrick Boerlin[4], Leonardo Susta[4]

1 Ontario Ministry of Agriculture, Food and Rural Affairs, Guelph, Ontario, Canada, 2 Department of Population Medicine, Ontario Veterinary College, University of Guelph, Guelph, Ontario, Canada, 3 Animal Health Laboratory, University of Guelph, Guelph, Ontario, Canada, 4 Department of Pathobiology, Ontario Veterinary College, University of Guelph, Guelph, Ontario, Canada

* cvarga@uoguelph.ca

**Data Availability Statement:** All relevant data are within the manuscript.

**Funding:** The surveillance project led by Dr. L. Susta, was funded by the OMAFRA-University of

## Abstract

Antimicrobial resistance in *Campylobacter*, common in poultry, is a global public health issue. The emergence and spread of antimicrobial resistant *Campylobacter* has been linked to the use of antimicrobials in food animals. Small poultry flocks are becoming increasingly popular not only as a source of food but also as pets, yet not all small flock owners are aware of proper antimicrobial use practices and safe food handling protocols. This trend could contribute to antimicrobial resistance. In order to determine the prevalence of antimicrobial resistance in *Campylobacter* in small poultry flocks, we analyzed data from birds that had been submitted to a diagnostic laboratory in Ontario between October 2015 and September 2017. A pooled cecal sample was obtained from each submission and cultured for *Campylobacter jejuni* and *Campylobacter coli*. Three isolates were recovered from each positive sample and tested for susceptibility to nine antimicrobials using a broth microdilution method. Overall, 176 isolates were recovered (141 chicken, 21 turkey, 6 duck, and 8 game bird). A high frequency of resistance to tetracycline was observed in the *C. jejuni* isolates from chickens (77%) and turkeys (100%), and in the *C. coli* isolates from turkeys (50%) and game birds (40%). *Campylobacter jejuni* isolates had higher odds of resistance to tetracycline (OR = 3.54, $P \leq 0.01$) compared to *C. coli* isolates. Overall, there was a low frequency of resistance to quinolones and a very low frequency of resistance to macrolides. Multidrug resistance was uncommon. The high prevalence of tetracycline resistance emphasizes the importance of prudent antimicrobial use in small flocks. Although low, the presence of resistance to macrolides and quinolones, which are used to treat campylobacteriosis in humans, highlights the need for proper food safety and infection control practices by small flock owners to prevent exposure to antimicrobial resistant *Campylobacter*.

Guelph Strategic Partnership (grant UofG 2015-2282), under the Disease Surveillance Plan, which was a joint federal-provincial Growing Forward 2 project. The antimicrobial susceptibility testing project led by Dr. C. Varga, was funded by the Ontario Agri-Food Innovation Alliance, under the Disease Surveillance Plan (Project #: 009098).

**Competing interests:** The authors have declared that no competing interests exist.

## Introduction

Small poultry flocks are becoming increasingly popular in Canada [1, 2]. Poultry are reservoirs of thermophilic *Campylobacter* species, especially *Campylobacter jejuni* and *Campylobacter coli*, and they shed the bacteria in their feces [3]. *Campylobacter* can be transmitted from backyard poultry to humans through activities that might expose a flock owner to their birds' feces, such as cleaning the birds' housing and removing soiled bedding, bird handling, petting, and kissing [4], or through handling and consumption of contaminated eggs and meat [3]. *Campylobacter* are a significant cause of enteric disease in humans [5–7]. Antimicrobial resistance (AMR) poses an additional risk [8, 9] because infections caused by antimicrobial resistant *Campylobacter* lead to longer hospitalizations, higher treatment failures, and increased morbidity and mortality [10].

The transmission of antimicrobial resistant *Campylobacter* from commercial broiler and free-range chickens to humans has been described previously [11]. In Canada, *Campylobacter* isolates from retail poultry meat were genetically linked to isolates from human clinical cases, and a high proportion of those isolates were resistant to tetracycline [12]. Moreover, other Canadian research studies and surveillance programs have reported a high prevalence of tetracycline and fluoroquinolone resistance among *Campylobacter* isolates from on-farm and abattoir chicken samples in Ontario, and from retail chicken meat samples in British Columbia, Saskatchewan, and Ontario [13–15]. A high prevalence of tetracycline and fluoroquinolone resistance in *Campylobacter* isolates from commercial poultry has also been reported in Poland [16], China [17], and Italy [18].

Antimicrobial use, including overuse and misuse in food animals, is considered a contributing factor for the selection and emergence of antimicrobial resistant enteric bacteria [19, 20]. Until recently, small flock owners in Canada could purchase antimicrobials over the counter without a veterinary prescription [21]. This might have favoured improper antimicrobial use.

There are few research studies on AMR in enteric bacteria from small flocks [22, 23]. To address this knowledge gap, we evaluated AMR patterns in *C. jejuni* and *C. coli* isolates from small flock chickens, turkeys, waterfowl, and game birds submitted to a veterinary diagnostic laboratory in Ontario.

## Methods

### Study design

Samples were obtained through a 2-year prospective surveillance study conducted in Ontario from October 2015 to September 2017 [24]. In brief, small flock owners, through their veterinarian, submitted cases suffering production issues, clinical illness, or mortality to the Animal Health Laboratory, University of Guelph. A submission consisted of not more than 5 sick and/or dead birds of one species from the same flock. Small poultry flocks were defined as flocks consisting of not more than 299 broiler chickens, 99 layer chickens, 49 turkeys, 300 waterfowl, or 300 game birds.

### Sample collection and *Campylobacter* isolation

*Campylobacter* isolation and antimicrobial susceptibility testing were performed at the Animal Health Laboratory in Guelph, Ontario. One pooled cecal sample was collected from each submission and cultured for *Campylobacter*. The cecal material was directly plated onto Campylobacter Blood Free media (Bio-Media Unlimited Ltd., Toronto, Ontario, Canada) and incubated in a microaerophilic environment at 37°C for 72 h. Presumptive *Campylobacter* colonies (i.e., yellowish-gray, translucent, round, 1–2 mm diameter, smooth to slightly mucoid)

were selected and identified using matrix-assisted laser desorption ionization time-of-flight mass spectrometry (Bruker Ltd., Billerica, Massachusetts, USA).

## Antimicrobial susceptibility testing and classification

Three isolates were selected from each positive sample and tested for susceptibility to nine antimicrobials (NARMS CAMPY plates) using an automated broth microdilution technique (Sensititre®; Trek Diagnostic Systems Inc., Westlake, Ohio, USA). The minimum inhibitory concentration interpretive standards of the Canadian Integrated Program for Antimicrobial Resistance Surveillance (for most antimicrobials) [15] or the Centers for Disease Control and Prevention (for telithromycin) [25] were used to classify *Campylobacter* isolates as susceptible or resistant (resistant plus intermediate). The antimicrobials, and their antimicrobial class, concentration range, and established susceptibility breakpoints are presented in Table 1. In addition, an isolate was defined as multidrug resistant if it was non-susceptible to one or more antimicrobials in $\geq 3$ different antimicrobial classes [26].

## Data analysis

All analyses were conducted using Stata 14.2 (Stata Corp, College Station, Texas, USA).

**Descriptive statistics.** Overall, and for each poultry species (chicken, turkey, other), prevalence estimates for resistance of *C. jejuni* and *C. coli* isolates to each of the nine tested antimicrobials were calculated by dividing the number of isolates resistant to an antimicrobial by the total number of isolates tested for the antimicrobial. An exact binomial 95% confidence interval was computed for each prevalence estimate.

In addition, to account for sample-level clustering, adjusted prevalence estimates were computed using a generalized estimating equation (GEE) with a binary outcome, logit-link function, and exchangeable correlation structure. The intercepts ($\beta_0$) obtained from null binomial models were used to calculate population-averaged prevalence estimates using the following formula [27]:

$$P = [1 + \exp(-\beta_0)]^{-1}.$$

To achieve model convergence, adjusted prevalence estimates were only calculated for *Campylobacter* isolated from chicken samples. The 95% confidence interval of the intercept was used for each adjusted prevalence estimate.

**Table 1. Antimicrobial classes, antimicrobial agents, concentration ranges, and susceptibility breakpoints for *Campylobacter* isolates.**

| Antimicrobial Class | Antimicrobial Agent | Concentration Range (μg/mL) | MIC Interpretive Standard (μg/mL)[A] | |
|---|---|---|---|---|
| | | | Susceptible | Resistant |
| Aminoglycosides | Gentamicin | 0.12–32 | $\leq 2$ | $\geq 4$ |
| Ketolides | Telithromycin | 0.015–8 | $\leq 4$ | $\geq 8$ |
| Lincosamides | Clindamycin | 0.03–16 | $\leq 2$ | $\geq 4$ |
| Macrolides | Azithromycin | 0.015–64 | $\leq 2$ | $\geq 4$ |
| | Erythromycin | 0.03–64 | $\leq 8$ | $\geq 16$ |
| Phenicols | Florfenicol | 0.03–64 | $\leq 4$ | N/A |
| Quinolones | Ciprofloxacin | 0.015–64 | $\leq 1$ | $\geq 2$ |
| | Nalidixic acid | 4–64 | $\leq 16$ | $\geq 32$ |
| Tetracyclines | Tetracycline | 0.06–64 | $\leq 4$ | $\geq 8$ |

[A]Isolates were tested for antimicrobial susceptibility using an automated broth microdilution technique (Sensititre®). Minimum inhibitory concentration (MIC) interpretive standards of the Canadian Integrated Program for Antimicrobial Resistance Surveillance (for most antimicrobials) [15] or the Centers for Disease Control and Prevention (for telithromycin) [25] were used to classify isolates as susceptible or resistant (resistant plus intermediate).

Further, prevalence estimates for multidrug resistance were calculated by dividing the number of multidrug resistant *Campylobacter* isolates by the total number of isolates tested.

**Cluster analyses.** To compare individual antimicrobials in terms of their similarity in *Campylobacter* resistance, cluster analysis, using the Jaccard binary similarity coefficient, was performed for *C. jejuni* and *C. coli* isolates from chicken samples. Cluster analysis was not performed for isolates from turkeys or other poultry species due to the small number of isolates. The number of isolates that are resistant to both antimicrobials, and the number that are resistant to one and susceptible to the other are utilized in the calculation of the coefficient. Dendrograms were created using the single-linkage clustering technique with the Jaccard distance. The Jaccard distance measures dissimilarity between antimicrobials and is calculated by subtracting the Jaccard similarity coefficient from one [28].

To explore relationships within the set of nine selected antimicrobials in terms of their similarity in *Campylobacter* resistance, multiple correspondence analysis, using the Burt method with principal normalization [29, 30], was conducted for *C. jejuni* and *C. coli* isolates from chicken samples. Multiple correspondence analysis was not conducted for isolates from turkeys or other poultry species due to the small number of isolates and lack of variation. Dimensions that explained at least two-thirds of the variation in the data were included for further analysis. Observation scores were calculated and plotted to visualize the distribution of antimicrobial resistance patterns along the first two dimensions.

**Logistic regression.** To identify differences in *Campylobacter* resistance between poultry species, logistic regression was used. Only antimicrobials for which $\geq$ 5% of the isolates were resistant were assessed. Therefore, 3 of 9 antimicrobials were analyzed: ciprofloxacin, nalidixic acid, and tetracycline. Three population-averaged models were fit for each antimicrobial using the GEE described previously. In these univariable models, the dependent variable represented the prevalence of resistance to the antimicrobial, while the independent dichotomous variable was poultry species (model 1: chickens compared to all the other poultry species; model 2: turkeys compared to all the other poultry species; model 3: ducks and game birds compared to all the other poultry species). One additional population-averaged model was fit for each antimicrobial (ciprofloxacin, nalidixic acid, tetracycline) to identify differences between species of *Campylobacter*. In these models, the dependent variable represented the prevalence of resistance to the antimicrobial, while the independent dichotomous variable was *Campylobacter* species (*C. jejuni* compared to *C. coli*). A *P*-value $\leq$ 0.05 on the Wald $\chi^2$ test implied a statistically significant association.

## Results

### Description of submissions

In total, the Animal Health Laboratory received 160 small flock submissions over the 2-year study period. The number of birds per submission ranged from 1–5 (median 1). Flock sizes ranged from 1–299 birds (median 25) and birds ranged from 6 days to 7 years of age (median 7 months). Most of the submissions were chickens (134), although a few were turkeys (10), ducks (8), and game birds (8) [24].

### Descriptive statistics

Of 158 submissions tested for *Campylobacter*, a total of 176 isolates were recovered: 141 isolates from chicken submissions (47 pooled samples, 3 isolates recovered from each pooled sample); 21 isolates from turkey submissions (7 pooled samples, 3 isolates recovered from each pooled sample); 6 isolates from duck submissions (2 pooled samples, 3 isolates recovered from each pooled sample); and 8 isolates from game bird (pheasant and quail) submissions (3

**Table 2. Percentage of *Campylobacter jejuni* and *Campylobacter coli* isolates from Ontario small poultry flocks that were resistant to nine selected antimicrobials, as determined by a broth microdilution technique, by poultry species.**

| Antimicrobial | | All poultry species | Chicken | | Turkey | | Other poultry species[A] | |
|---|---|---|---|---|---|---|---|---|
| | | *Campylobacter* (N = 176)[B] | *C. jejuni* (N = 77) | *C. coli* (N = 64) | *C. jejuni* (N = 3) | *C. coli* (N = 18) | *C. jejuni* (N = 6) | *C. coli* (N = 5) |
| Class | Agent[C] | %[D] [CI][E] | % [CI] | % [CI] | % [CI] | % [CI] | % [CI] | % [CI] |
| Aminoglycosides | GEN | 0.57 [0.01–3.12] | 0 | 1.56 [0.04–8.40] | 0 | 0 | 0 | 0 |
| Ketolides | TEL | 3.98 [1.61–8.02] | 0 | 10.94 [4.51–21.25] | 0 | 0 | 0 | 0 |
| Lincosamides | CLI | 3.98 [1.61–8.02] | 0 | 10.94 [4.51–21.25] | 0 | 0 | 0 | 0 |
| Macrolides | AZT | 4.55 [1.98–8.76] | 1.30 [0.04–7.02] | 10.94 [4.51–21.25] | 0 | 0 | 0 | 0 |
| | ERY | 3.98 [1.61–8.02] | 0 | 10.94 [4.51–21.25] | 0 | 0 | 0 | 0 |
| Phenicols | FLO | 0.57 [0.01–3.12] | 0 | 1.56 [0.04–8.40] | 0 | 0 | 0 | 0 |
| Quinolones | CIP | 8.52 [4.84–13.67] | 3.90 [0.81–10.97] | 9.38 [3.52–19.30] | 0 | 16.67 [3.58–41.42] | 50.00 [11.81–88.19] | 0 |
| | NAL | 7.39 [3.99–12.30] | 3.90 [0.81–10.97] | 9.38 [3.52–19.30] | 0 | 16.67 [3.58–41.42] | 0 | 20.00 [0.50–71.64] |
| Tetracyclines | TET | 56.25 [48.58–63.70] | 76.62 [65.59–85.52] | 35.94 [24.32–48.90] | 100 [29.24–100] | 50.00 [26.02–73.98] | 0 | 40.00 [5.27–85.34] |

[A]Waterfowl (ducks) and game birds (pheasant and quail).

[B]Antimicrobial susceptibility testing was conducted on all 176 isolates, including the three game bird isolates that were not speciated.

[C]AZT = azithromycin; CIP = ciprofloxacin; CLI = clindamycin; ERY = erythromycin; FLO = florfenicol; GEN = gentamicin; NAL = nalidixic acid; TEL = telithromycin; TET = tetracycline.

[D]Percentage of isolates resistant to the antimicrobial. Prevalence estimates were calculated by dividing the number of isolates resistant to an antimicrobial by the total number of isolates tested for the antimicrobial.

[E]CI = Exact binomial 95% confidence interval for the percentage of isolates resistant to the antimicrobial.

pooled samples in total; 3 isolates recovered from each of 2 pooled samples and 2 isolates recovered from 1 of the pooled samples). Of 176 isolates, 86 were *C. jejuni* (77 chicken, 3 turkey, 6 duck), 87 were *C. coli* (64 chicken, 18 turkey, and 5 game bird), and 3 were not speciated (3 game bird). Of the 176 isolates, 33.3% of the chicken (47/141), 42.9% of the turkey (9/21), and 42.9% of the other poultry species (6/14) were pan-susceptible.

Overall (i.e., all poultry species and all *Campylobacter* spp. combined), at the isolate-level, there was a high prevalence of resistance (≥ 40% of isolates) to tetracycline, a low prevalence of resistance (5–14% of isolates) to ciprofloxacin and nalidixic acid, and a very low prevalence of resistance (< 5% of isolates) to gentamicin, telithromycin, clindamycin, azithromycin, erythromycin, and florfenicol (Table 2). In the chicken *C. jejuni* isolates, there was a high frequency of resistance to tetracycline. In the chicken *C. coli* isolates, there was a moderate frequency of resistance (15–39% of isolates) to tetracycline. In the turkey *C. jejuni* isolates, there was a high frequency of resistance to tetracycline. In the turkey *C. coli* isolates, there was a high frequency of resistance to tetracycline, and a moderate frequency of resistance to ciprofloxacin and nalidixic acid. In the *C. jejuni* isolates of other poultry species, there was a high frequency of resistance to ciprofloxacin. In the *C. coli* isolates of other poultry species, there was a high frequency of resistance to tetracycline and a moderate frequency of resistance to nalidixic acid. There was no resistance detected for most of the antimicrobials (6 of 9) in the turkey, waterfowl, and game bird isolates.

The adjusted prevalence estimates of AMR in *Campylobacter* spp. isolated from chicken samples are presented in Table 3. At the sample-level, there was a high prevalence of resistance

**Table 3. Adjusted prevalence of antimicrobial resistance, accounting for sample-level clustering, in *Campylobacter* isolates from chicken cecal samples from small poultry flocks in Ontario between October 2015 and September 2017 using population-averaged logistic regression models.**

| Antimicrobial | | *Campylobacter* spp. (N = 141) |
|---|---|---|
| Class | Agent | Percentage resistant [95% Confidence Interval] |
| Aminoglycosides | Gentamicin | 0.71 [0.10–4.81] |
| Ketolides | Telithromycin | 5.28 [1.67–15.49] |
| Lincosamides | Clindamycin | 5.28 [1.67–15.49] |
| Macrolides | Azithromycin | 6.02 [2.14–15.79] |
| | Erythromycin | 5.28 [1.67–15.49] |
| Phenicols | Florfenicol | 0.71 [0.10–4.81] |
| Quinolones | Ciprofloxacin | 6.81 [2.34–18.18] |
| | Nalidixic acid | 6.81 [2.34–18.18] |
| Tetracyclines | Tetracycline | 58.69 [44.73–71.39] |

to tetracycline, a low prevalence of resistance to telithromycin, clindamycin, azithromycin, erythromycin, ciprofloxacin, and nalidixic acid, and a very low prevalence of resistance to gentamicin and florfenicol.

The most common antimicrobial resistance patterns in *Campylobacter* spp. isolated from chicken samples are presented in Table 4. Multidrug resistance was detected in 4.26% (6/141) of isolates (all *C. coli*) from chicken samples; however, it was not detected in *C. jejuni* chicken isolates, or in turkey, duck, or game bird *Campylobacter* isolates.

## Cluster analyses

Single-linkage clustering dendrograms with Jaccard distances for *C. jejuni* and *C. coli* isolates from chicken samples are presented in Figs 1 and 2, respectively. A low, non-zero dissimilarity measure implies that a relatively high proportion of isolates were resistant to both antimicrobials, whereas a high dissimilarity measure implies that relatively few isolates were resistant to both antimicrobials, and a dissimilarity measure of zero implies that all isolates were susceptible to both antimicrobials. For *C. jejuni*, there was one cluster of isolates susceptible to ciprofloxacin and nalidixic acid, and a second cluster of isolates susceptible to gentamicin,

**Table 4. Most common antimicrobial resistance patterns of *Campylobacter* isolates (N = 141) from chicken cecal samples from small poultry flocks in Ontario between October 2015 and September 2017.**

| Antimicrobial resistance pattern[A] | Number of antimicrobial classes in pattern (multidrug resistant)[B] | n (%)[C] |
|---|---|---|
| TET | 1 (no) | 77 (54.61) |
| AZT-CLI-ERY-TEL | 3 (yes) | 6 (4.26) |
| CIP-NAL | 1 (no) | 5 (3.55) |
| CIP-NAL-TET | 2 (no) | 3 (2.13) |

[A]Resistance to nine selected antimicrobials (including gentamicin and florfenicol) as determined by a broth microdilution technique. AZT = azithromycin; CIP = ciprofloxacin; CLI = clindamycin; ERY = erythromycin; NAL = nalidixic acid; TEL = telithromycin; TET = tetracycline.

[B]An isolate was defined as multidrug resistant if it was non-susceptible to one or more antimicrobials in ≥ 3 different antimicrobial classes (Ketolides: TEL; Lincosamides: CLI; Macrolides: AZT, ERY; Quinolones: CIP, NAL; Tetracyclines: TET).

[C]Number and percentage of isolates with each antimicrobial resistance pattern. Only patterns with ≥ 3 isolates are shown.

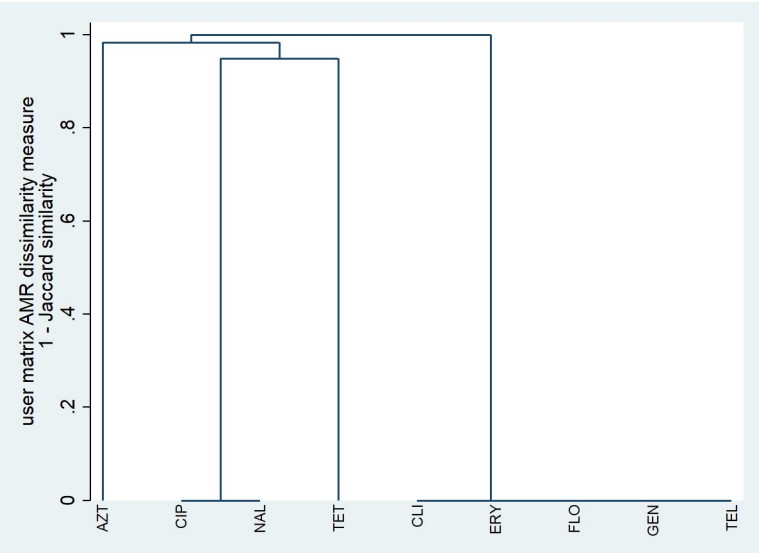

**Fig 1. Single-linkage clustering dendrogram for resistance to nine antimicrobials in *Campylobacter jejuni* isolates from chicken cecal samples from small poultry flocks in Ontario between October 2015 and September 2017 (n = 77).** AZT = azithromycin; CIP = ciprofloxacin; CLI = clindamycin; ERY = erythromycin; FLO = florfenicol; GEN = gentamicin; NAL = nalidixic acid; TEL = telithromycin; TET = tetracycline.

telithromycin, clindamycin, erythromycin, and florfenicol. For *C. coli*, a relatively high proportion (i.e., a cluster) of isolates were resistant to telithromycin, clindamycin, azithromycin, and erythromycin. In addition, there was one cluster of isolates susceptible to ciprofloxacin and nalidixic acid, a second cluster susceptible to gentamicin and florfenicol, and a third cluster susceptible to clindamycin, azithromycin, and erythromycin.

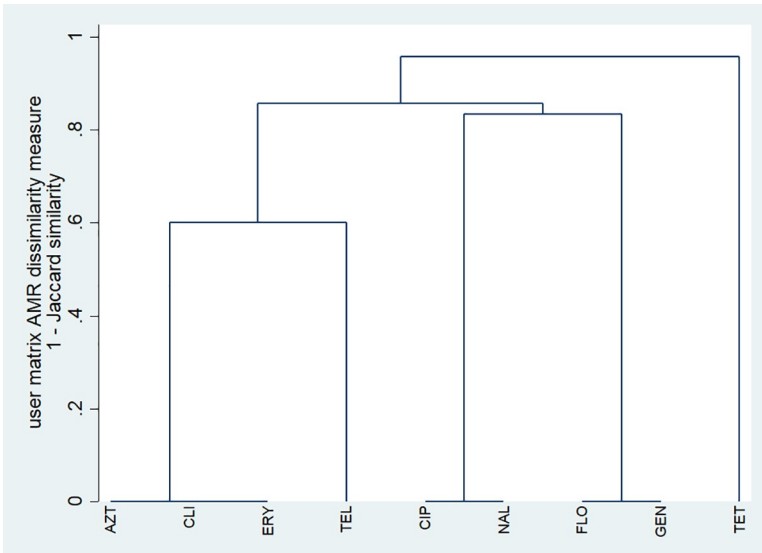

**Fig 2. Single-linkage clustering dendrogram for resistance to nine antimicrobials in *Campylobacter coli* isolates from chicken cecal samples from small poultry flocks in Ontario between October 2015 and September 2017 (n = 64).** AZT = azithromycin; CIP = ciprofloxacin; CLI = clindamycin; ERY = erythromycin; FLO = florfenicol; GEN = gentamicin; NAL = nalidixic acid; TEL = telithromycin; TET = tetracycline.

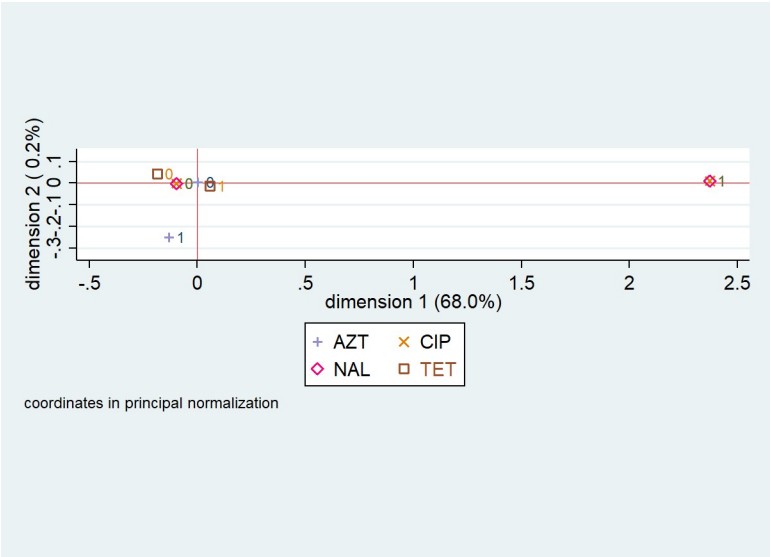

**Fig 3. Multiple correspondence analysis coordinate plot displaying the presence (1) and absence (0) of resistance to four antimicrobials in *Campylobacter jejuni* isolates from chicken cecal samples from small poultry flocks in Ontario between October 2015 and September 2017 for the first two dimensions (n = 77).** AZT = azithromycin; CIP = ciprofloxacin; NAL = nalidixic acid; TET = tetracycline. Five antimicrobials (gentamicin, telithromycin, clindamycin, erythromycin, and florfenicol) were omitted from the analysis because they completely predicted the presence or absence of resistance (i.e., there was no variation).

Multiple correspondence analysis coordinate plots for the first two dimensions for resistance in *C. jejuni* and *C. coli* isolates from chicken samples are presented in Figs 3 and 4, respectively. Five antimicrobials (gentamicin, telithromycin, clindamycin, erythromycin, and

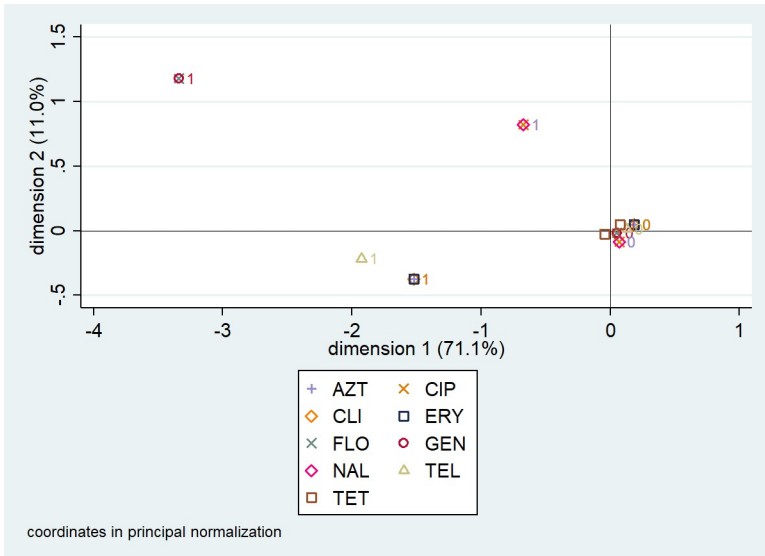

**Fig 4. Multiple correspondence analysis coordinate plot displaying the presence (1) and absence (0) of resistance to nine antimicrobials in *Campylobacter coli* isolates from chicken cecal samples from small poultry flocks in Ontario between October 2015 and September 2017 for the first two dimensions (n = 64).** AZT = azithromycin; CIP = ciprofloxacin; CLI = clindamycin; ERY = erythromycin; FLO = florfenicol; GEN = gentamicin; NAL = nalidixic acid; TEL = telithromycin; TET = tetracycline.

florfenicol) were omitted from the *C. jejuni* analysis because they completely predicted the presence or absence of resistance (i.e., there was no variation). For *C. jejuni*, the first two dimensions explained 68.2% of the variation in antimicrobial resistance (Fig 3). When observation scores were plotted along dimensions 1 and 2, it was observed that the antimicrobial susceptibility testing results (resistant and susceptible denoted as 1 and 0, respectively) for ciprofloxacin and nalidixic acid formed a pattern (i.e., were clustered together). For *C. coli*, the first two dimensions explained 82.1% of the variation in antimicrobial resistance (Fig 4). Three patterns were identified: the antimicrobial susceptibility testing results for ciprofloxacin and nalidixic acid clustered together; results for gentamicin and florfenicol clustered together; and results for azithromycin, clindamycin, and erythromycin clustered together.

### Logistic regression

Poultry species were not significantly associated with the prevalence of resistance to individual antimicrobials among *Campylobacter* isolates. The odds of resistance to tetracycline were significantly higher in *C. jejuni* isolates (OR = 3.54, 95% CI: 2.00–6.26, $P \leq 0.01$) compared to *C. coli* isolates.

### Discussion

In this study, we assessed the AMR patterns in *C. jejuni* and *C. coli* isolates from Ontario small poultry flocks that were submitted by veterinarians to a diagnostic laboratory because of mortality, morbidity, or production problems. A high proportion of the *Campylobacter* isolates (56%) were resistant to tetracycline, particularly the *C. jejuni* isolates. This finding is in agreement with a study in Kenya where a high proportion of *C. jejuni* and *C. coli* isolates from small scale and backyard chicken flocks were resistant to tetracycline [31], and with a study in Finland where a high proportion of *C. jejuni* isolates from backyard chicken flocks were resistant to tetracycline [32]. High frequencies of tetracycline resistance have also been identified in *Campylobacter* isolates (45–96%) from retail meat and commercial poultry flock samples in research studies in Canada [12], China [17], Poland [16], and Italy [18].

It is recognized that antimicrobial use, including overuse and misuse in food animals, is a contributing factor to AMR emergence in commensal and zoonotic enteric bacteria [33, 34]. Previous studies have shown that tetracycline resistance in *Campylobacter* is generally determined by a plasmid-encoded *tet*(O) gene [35], and that this resistance gene can be transferred horizontally between *C. jejuni* and *C. coli* isolates in the intestinal tract of food animals and humans [36, 37]. Tetracycline is commonly used to treat bacterial diseases of poultry and it was readily available to small flock owners over the counter at livestock medicine outlets during the period of study. In our study, 61% of the flock owners administered medication to their flock within the last 12 months, and 61% of those gave antibiotics, including tetracycline, penicillin, and tylosin (unpublished data). This easy access and potential preferential use could have been a source of selection pressure, resulting in the comparatively higher prevalence of tetracycline resistance in our study than in Canadian commercial chicken flocks (20%) [15].

To reduce the development and spread of AMR in enteric bacteria of food animals, an updated antimicrobial use regulation was implemented in Canada on December 1, 2018 [21]. Under this regulation, a veterinary prescription is required [21] for all medically important antimicrobials in human medicine [38]. As our study was conducted before the updated regulation came into effect, our findings will provide a benchmark for AMR in *Campylobacter* isolates from small poultry flocks.

Our single-linkage clustering and multiple correspondence analyses revealed a high degree of relatedness between resistance to ciprofloxacin (a fluoroquinolone) and nalidixic acid (a

quinolone) in the *C. jejuni* and *C. coli* isolates from chicken samples. Fluoroquinolone-resistance in *Campylobacter* is generally mediated by mutations in the gyrA gene at the Thr-86 position [16]; these mutations lead to resistance to both ciprofloxacin and nalidixic acid [16], and resistant strains may have a competitive advantage over susceptible strains in colonizing the intestinal tract of poultry, even without antimicrobial use selection pressure [39]. Therefore, although the prevalence of quinolone resistance was low, small flock owners should follow good husbandry, biosecurity, and food safety practices given the health risk associated with fluoroquinolone-resistant *Campylobacter* infection acquired from consumption of contaminated meat [12] or direct contact with live poultry that are actively shedding the bacteria [40]. On the other hand, the lack of relatedness between resistance to the quinolones and other antimicrobial classes is an encouraging finding.

In our study, although we could not assess this statistically, a relatively higher number of *C. coli* isolates from chicken samples were resistant to azithromycin and erythromycin (macrolides) compared to *C. jejuni* isolates. Macrolides are first choice antibiotics for the treatment of campylobacteriosis in humans [41]. The emergence of macrolide resistance is supported by substitutions in the 23S rRNA gene, specifically A2075G, and less frequently A2074C/G [42]. Interestingly, this mutation in *C. jejuni* is associated with a decreased ability to colonize chickens; however, this reduced colonization ability has not been observed in *C. coli* [42], which might explain the lower frequency of macrolide resistance in the *C. jejuni* isolates.

Although multidrug resistance was uncommon, the azithromycin—clindamycin—erythromycin—telithromycin pattern was identified in our study. Despite belonging to three different antimicrobial classes (macrolides, lincosamides, and ketolides) and having chemically distinct structures, these four antimicrobials are related molecules that have similar modes of antibacterial action and a single mechanism that encodes resistance for all at once [43–45].

Overall, we found a very low frequency of resistance to ketolides and aminoglycosides, which are categorized in Canada as being of very high and high importance in human medicine, respectively [38]. Further, only a few *C. coli* chicken isolates were multidrug resistant, and a moderate to high percentage of isolates were pan-susceptible, which is a promising finding from a human health perspective. The high degree of relatedness between gentamicin (an aminoglycoside) and florfenicol (a phenicol) in the isolates from chicken samples was unexpected and could be an incidental finding given the very low prevalence of resistance to these antimicrobials.

Interestingly, there were no differences between poultry species in terms of resistance to individual antimicrobials. However, because the majority of the isolates in our study were from chicken samples, future studies that include a larger number of isolates from other poultry species are needed to identify factors that influence the development of AMR in *Campylobacter* in small flocks.

Limitations of our study should be considered when interpreting our findings. Our study might overestimate the frequency of AMR in *Campylobacter* isolates because isolates were obtained from diagnostic submissions of diseased birds that might have been treated with antimicrobials. Also, samples were not obtained randomly, and submissions from areas closer to the diagnostic laboratory were overrepresented [24].

In conclusion, a high proportion of isolates were resistant to tetracycline, an antimicrobial commonly used to treat bacterial diseases of poultry, which emphasizes the importance of prudent antimicrobial use in small flocks. Although low, the presence of resistance to macrolides and quinolones, which are used to treat campylobacteriosis in humans, highlights the need for proper food safety and infection control practices by small flock owners to prevent the transmission of antimicrobial resistant *Campylobacter* through consumption of contaminated poultry products or direct contact with infected birds. The very low prevalence of resistance to

ketolides and aminoglycosides is an encouraging finding from a public health standpoint. As our study was conducted before the antimicrobial use regulation was updated, the results can be used by governmental agencies and researchers as a benchmark to measure changes in AMR patterns in *Campylobacter* isolates of small poultry flocks.

## Acknowledgments

The authors thank the veterinarians and small flock owners who submitted birds and participated in this study. We also thank the laboratory staff of the Animal Health Laboratory for coordinating and processing all submissions and performing antimicrobial susceptibility testing.

## Author Contributions

**Conceptualization:** Csaba Varga, Michele T. Guerin, Marina L. Brash, Leonardo Susta.

**Data curation:** Csaba Varga, Michele T. Guerin.

**Formal analysis:** Csaba Varga.

**Funding acquisition:** Csaba Varga, Leonardo Susta.

**Investigation:** Csaba Varga, Michele T. Guerin, Marina L. Brash, Patrick Boerlin, Leonardo Susta.

**Methodology:** Csaba Varga, Michele T. Guerin, Marina L. Brash, Patrick Boerlin, Leonardo Susta.

**Project administration:** Csaba Varga, Durda Slavic.

**Resources:** Csaba Varga, Marina L. Brash, Durda Slavic, Leonardo Susta.

**Software:** Csaba Varga.

**Supervision:** Csaba Varga.

**Validation:** Csaba Varga.

**Visualization:** Csaba Varga.

**Writing – original draft:** Csaba Varga.

**Writing – review & editing:** Csaba Varga.

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
