## [Decision Letter · Decision Letter 0]

10 Jul 2019

PONE-D-19-16109

Antimicrobial resistance in Campylobacter jejuni and Campylobacter coli isolated from small poultry flocks in Ontario, Canada: A two-year surveillance study

PLOS ONE

Dear Dr. Varga,

Thank you for submitting your manuscript to PLOS ONE. After careful consideration, we feel that it has merit but does not fully meet PLOS ONE’s publication criteria as it currently stands. Therefore, we invite you to submit a revised version of the manuscript that addresses the points raised during the review process.

Two reviewers have commented on the manuscript. Some minor revisions are required, particularly related to medium used and clarifications on number of isolates. Please address all points raised by the referees.

We would appreciate receiving your revised manuscript by Aug 24 2019 11:59PM. To enhance the reproducibility of your results, we recommend that if applicable you deposit your laboratory protocols in protocols.io, where a protocol can be assigned its own identifier (DOI) such that it can be cited independently in the future. For instructions see: http://journals.plos.org/plosone/s/submission-guidelines#loc-laboratory-protocols

We look forward to receiving your revised manuscript.

Kind regards,

Iddya Karunasagar

Academic Editor

PLOS ONE

Journal Requirements:

Additional Editor Comments:

Two reviewers have commented on the manuscript. Some minor revisions are required, particularly related to medium used and clarifications on number of isolates. Please address all points raised by the referees.

Reviewers' comments:

Reviewer's Responses to Questions

**Comments to the Author**

1. Is the manuscript technically sound, and do the data support the conclusions?

Reviewer #1: Yes

Reviewer #2: Yes

2. Has the statistical analysis been performed appropriately and rigorously? 

Reviewer #1: Yes

Reviewer #2: Yes

3. Have the authors made all data underlying the findings in their manuscript fully available?

Reviewer #1: Yes

Reviewer #2: Yes

4. Is the manuscript presented in an intelligible fashion and written in standard English?

Reviewer #1: Yes

Reviewer #2: Yes

5. Review Comments to the Author

Reviewer #1: The work presented in this manuscript is very relevant to our understanding of AMR in Campylobacter. The study is technically sound and the statistical analysis appropriate for the work. The study is particularly valuable as it was conducted before a change of regulation, which has now been implemented, and will represent a baseline for future work in the area. The authors have not overstated their conclusions and categorically indicate the limitations of the study.

Reviewer #2: The authors’ describe antimicrobial resistance (AMR) in Campylobacter from small poultry flocks in Ontario, Canada. The manuscript is well written. With a few revisions, it should be acceptable for publication.

Specific comments:

Line 49: This sentence need a reference(s).

Lines 78-79: I searched for Campy-Charcoal media at Bio-Media, but could not find it. Please verify the media you used and add this to the revised manuscript.

Line 82: What does a presumptive Campylobacter colony look like? Please indicate this in the revised manuscript.

Lines 153-162: The numbers presented here are confusing and don’t add up, especially on line 157 with the game birds. Can this be rewritten to clarify the number of isolates?

Lines 230-233: Most thermophillic Campylobacter are naturally competent. However, Campylobacter restricts most foreign DNA based on methylation (https://www.ncbi.nlm.nih.gov/pubmed/28855338). I don’t agree tetO can be transferred and expressed in Campylobacter from commensals. Please think about this sentence and revise.

6. PLOS authors have the option to publish the peer review history of their article (what does this mean?). If published, this will include your full peer review and any attached files.

Reviewer #1: No

Reviewer #2: No

---

## [Author Response · Author response to Decision Letter 0]

1 Aug 2019

PONE-D-19-16109

Antimicrobial resistance in Campylobacter jejuni and Campylobacter coli isolated from small poultry flocks in Ontario, Canada: A two-year surveillance study

Reviewer #1: The work presented in this manuscript is very relevant to our understanding of AMR in Campylobacter. The study is technically sound and the statistical analysis appropriate for the work. The study is particularly valuable as it was conducted before a change of regulation, which has now been implemented, and will represent a baseline for future work in the area. The authors have not overstated their conclusions and categorically indicate the limitations of the study.

Reviewer #2: The authors’ describe antimicrobial resistance (AMR) in Campylobacter from small poultry flocks in Ontario, Canada. The manuscript is well written. With a few revisions, it should be acceptable for publication.

Specific comments:

Line 49: This sentence needs a reference(s).

A reference (de Vries et al., ~11) was added as requested (Lines 49 and 360 – 363). Note that the citations and Reference List had to be updated due to the addition of this reference.

We also rewrote the sentence to make it more specific: “The transmission of antimicrobial resistant Campylobacter from commercial broiler and free-range chickens to humans has been described previously [11]” (Lines 48-49).

Lines 78-79: I searched for Campy-Charcoal media at Bio-Media, but could not find it. Please verify the media you used and add this to the revised manuscript.

The information on Campylobacter media was verified and updated as requested:

“The cecal material was directly plated onto Campylobacter Blood Free media (Bio-Media Unlimited Ltd., Toronto, Ontario, Canada)” (Lines 78-79).

Please find the list of products available at Bio Media, as a reference:

https://www.bio-media.ca/Bio_Media/Bio_Media_Products.htm

Line 82: What does a presumptive Campylobacter colony look like? Please indicate this in the revised manuscript.

The description of a presumptive Campylobacter colony was added as requested:

“Presumptive Campylobacter colonies (i.e., yellowish-gray, translucent, round, 1-2 mm diameter, smooth to slightly mucoid) were selected (Lines 80-82).

Lines 153-162: The numbers presented here are confusing and don’t add up, especially on line 157 with the game birds. Can this be rewritten to clarify the number of isolates?

We verified that the numbers are correct and rewrote the sentence for clarification:

“Of 158 submissions tested for Campylobacter, a total of 176 isolates were recovered: 141 isolates from chicken submissions (47 pooled samples, 3 isolates recovered from each pooled sample); 21 isolates from turkey submissions (7 pooled samples, 3 isolates recovered from each pooled sample); 6 isolates from duck submissions (2 pooled samples, 3 isolates recovered from each pooled sample); and 8 isolates from game bird (pheasant and quail) submissions (3 pooled samples in total; 3 isolates recovered from each of 2 pooled samples and 2 isolates recovered from 1 of the pooled samples).” (Lines 154-160)

Lines 230-233: Most thermophillic Campylobacter are naturally competent. However, Campylobacter restricts most foreign DNA based on methylation (https://www.ncbi.nlm.nih.gov/pubmed/28855338). I don’t agree tetO can be transferred and expressed in Campylobacter from commensals. Please think about this sentence and revise.

The reference (Stevenson et al., ~37) was replaced with the reference (Kim et al., ~37) (Line 235) and the sentence was revised as requested:

“Previous studies have shown that tetracycline resistance in Campylobacter is generally determined by a plasmid-encoded tet(O) gene [35], and that this resistance gene can be transferred horizontally between C. jejuni and C. coli isolates in the intestinal tract of food animals and humans [36, 37]”. (Lines 231-235).

---

## [Editor Report · Decision Letter 1]

7 Aug 2019

Antimicrobial resistance in Campylobacter jejuni and Campylobacter coli isolated from small poultry flocks in Ontario, Canada: A two-year surveillance study

PONE-D-19-16109R1

Dear Dr. Varga,

We are pleased to inform you that your manuscript has been judged scientifically suitable for publication and will be formally accepted for publication once it complies with all outstanding technical requirements.

With kind regards,

Iddya Karunasagar

Academic Editor

PLOS ONE

Additional Editor Comments (optional):

The revisions made are satisfactory
---

## [Editor Report · Acceptance letter]

20 Aug 2019

PONE-D-19-16109R1 

Antimicrobial resistance in *Campylobacter jejuni* and *Campylobacter coli* isolated from small poultry flocks in Ontario, Canada: A two-year surveillance study 

Dear Dr. Varga:

I am pleased to inform you that your manuscript has been deemed suitable for publication in PLOS ONE. Congratulations! Your manuscript is now with our production department. 

With kind regards,

on behalf of

Dr. Iddya Karunasagar 

Academic Editor

PLOS ONE